# Modeling and Compensation of Positioning Error in Micromanipulation

**DOI:** 10.3390/mi14040779

**Published:** 2023-03-30

**Authors:** Miao Hao, Bin Yang, Changhai Ru, Chunfeng Yue, Zongjie Huang, Rongan Zhai, Yu Sun, Yong Wang, Changsheng Dai

**Affiliations:** 1School of Mechanical and Electrical Engineering, Soochow University, Suzhou 215137, China; 2The Reproductive Medicine Centre, The First Affiliated Hospital of Suzhou University, Suzhou 215031, China; 3School of Electronic and Information Engineering, Suzhou University of Science and Technology, Suzhou 215009, China; 4Suzhou Boundless Medical Technology Co., Ltd., Suzhou 215163, China; 5School of Mechatronic Engineering and Automation, Shanghai University, Shanghai 200444, China; 6Department of Mechanical and Industrial Engineering, University of Toronto, Toronto, ON M5S 3G8, Canada; 7School of Mechanical Engineering, Dalian University of Technology, Dalian 116081, China; daichangsheng@gmail.com

**Keywords:** micromanipulation platform, systematic error, image stitching, error compensation model

## Abstract

In order to improve the positioning accuracy of the micromanipulation system, a comprehensive error model is first established to take into account the microscope nonlinear imaging distortion, camera installation error, and the mechanical displacement error of the motorized stage. A novel error compensation method is then proposed with distortion compensation coefficients obtained by the Levenberg–Marquardt optimization algorithm combined with the deduced nonlinear imaging model. The compensation coefficients for camera installation error and mechanical displacement error are derived from the rigid-body translation technique and image stitching algorithm. To validate the error compensation model, single shot and cumulative error tests were designed. The experimental results show that after the error compensation, the displacement errors were controlled within 0.25 μm when moving in a single direction and within 0.02 μm per 1000 μm when moving in multiple directions.

## 1. Introduction

With the rapid development of modern biomedical technology, micromanipulation techniques have obtained widespread application, including nucleus transfer, microinjection, microdissection, embryo transfer, etc. However, accurate positioning plays a vital role in the micromanipulation [1,2,3]. Micro positioning technology uses image processing algorithms to visually locate the cells and microorganisms, converts image coordinates into object coordinates, and sends the converted coordinate information to the end effector for corresponding micro positioning [4]. Therefore, it is of great significance to study the systematic errors and compensation methods of microscopic vision systems. In order to improve the precision of micro-positioning, digital image correlation (DIC) is usually used to compensate the coordinate conversion error between the vision module of the micro-operating system and the end-effector [5,6,7,8,9,10]. For example, Lee [11] proposed a vision based high precision self-calibration method by the use of a designed chess board for converting the position relationship between the end-effector and the camera without operator intervention in the calibration process. At high magnifications, however, the optical microscope imaging cannot strictly meet the pinhole imaging model due to lens distortion, which will deteriorate the measurement of DIC [12,13,14]. Normally, the rigid-body translation technique is used for eliminating the distortion of the microscope camera [15,16,17,18,19]. Koide [20] describes a conversion technique based on reprojection error minimization by directly taking images of the calibration pattern. Malti [21] optimized the distortion parameters, the camera intrinsics and the hand–eye transform with epipolar constraints. Although the above method can effectively solve the camera distortion problem, it ignores the mechanical errors arising from the installation error of the camera and the mechanical displacement error. It can be seen that neither the rigid body translation method nor the DIC method can eliminate camera distortion errors and mechanical errors at the same time. In addition, in the current research, there is no comprehensive method to compensate these errors simultaneously.

The research object of this paper is a micromanipulation system based on machine vision, in which the camera and micromanipulation platform are the key components of positioning. In order to improve the positioning accuracy, not only the camera distortion error must be considered, but also the analysis and calculation of mechanical errors such as camera installation error and the mechanical displacement error of the micromanipulation platform. However, these are rarely discussed together in former works. Therefore, this paper proposes a novel error compensation model that unites the camera distortion errors compensation and the mechanical error compensation. The influences and derivation of the above errors are studied first. Then, a novel error compensation model is established to comprehensively take into account the microscope nonlinear imaging distortion, camera installation error, and the mechanical displacement error. Finally, to validate the error compensation model, single shot error tests and cumulative error tests were designed. The experimental results show that the proposed method is not only easy to implement, but also leads to accurate positioning.

## 2. System Overview

As shown in Figure 1, the micromanipulation system consists of a camera (scA1300-32 gm, Basler, Inc., Lubeck, Germany) which captures at 30 Hz in real time, a standard inverted microscope (Nikon Ti, Nikon, Inc., Tokyo, Japan), and a micromanipulation platform with a motorized X-Y stage (travel range of 75 mm along both axis, MLS-1740, Micro-Nano Automation Institute Co., Ltd., Suzhou City, China).

## 3. Derivation of System Errors

As the operating time of the micromanipulation system increases, the cumulative error of the system has a great impact on the accuracy of the micropositioning, and even leads to its failure. As shown in Figure 2, when the motorized stage is moved from point A to point C, due to the system errors, the actual displacement path of the platform deviates from the predetermined displacement path.

### 3.1. Image Model Error

The microscope imaging model will affect the positioning accuracy. At present, the commonly used imaging models are pinhole models or parallel projection models. However, the basic assumption of these imaging models is linearity, and the lens group of a microscope is susceptible to handling, shaking and other factors, causing distortion and non-linearity. Image distortion is caused by the manufacturing and assembly errors of the optical system, which will lead to the transform error while transforming between image coordinate and physical coordinate during micromanipulation [22]. In this paper, the microscope is equipped with a standard camera interface for installing a camera. There is a dedicated optical path inside the microscope for imaging, as shown in Figure 3a, which presents the composition diagram of the vision system. For a micro-vision system equipped with an infinity-corrected optical path, its basic geometric imaging model can be simplified as shown in Figure 3b.

Assume that there is a point in the physical coordinate system whose coordinate is *P = (X, Y, *0*)^T^*, and after microscope imaging, its corresponding coordinate in the image coordinate system is *p = (u, v)^T^*. If the image distortion is ignored, the relationship between *P* and *p* is obtained according to the geometric principle of 2D photography:(1)$$\lambda \tilde{p}=H\tilde{P}=A[R_{12}t]\tilde{P}$$
where λ represents any scale factor.

*H* represents the homography matrix of the imaging model.

$\tilde{p}$ and $\tilde{P}$ represent the homogeneous coordinates in the physical coordinate system and image coordinate system without considering the Z component, respectively.

*A* represents the internal parameter matrix $A=\left[\begin{array}{ccc} f_{x} & 0 & u_{0}\\ 0 & f_{y} & v_{0}\\ 0 & 0 & 1 \end{array}\right]$.

*R*_12_ represents the first two columns of the 3 × 3 external parameter rotation matrix *R*. $R=\left[R_{12}r_{3}\right]=\left[r_{1}r_{2}r_{3}\right]$.

*t* represents the translation vector in the external parameter matrix. $t=\left[t_{1}t_{2}t_{3}\right]$.

Assuming that *L* and *l* represent a pair of lines in the physical coordinate system and image coordinate system, respectively. According to the contravariant property of point-line mapping, the relationship between its homogeneous coordinates can be expressed as:(2)$$\lambda \tilde{l}=H^{-1}\tilde{L}$$

However, due to the existence of lens distortion, the linear model in Equations (1) and (2) does not apply. The relationship between theoretical projection coordinates $p(\tilde{u},\tilde{v})$ and actual projection coordinates $p^{\prime }(u,v)$ in the image coordinate system is non -linear, which can be described as:(3)$$\left[\begin{array}{c} \tilde{u}\\ \tilde{v} \end{array}\right]=\left[\begin{array}{c} u+k(u-u_{0})\left[{\left(u-u_{0}\right)}^{2}+{\left(v-v_{0}\right)}^{2}\right]\\ v+k(v-v_{0})\left[{\left(u-u_{0}\right)}^{2}+{\left(v-v_{0}\right)}^{2}\right] \end{array}\right]$$
where *k* is a second-order distortion factor.

The nonlinear imaging model formula can be obtained as:(4)$$\lambda \left[\begin{array}{c} u+k(u-u_{0})\left[{\left(u-u_{0}\right)}^{2}+{\left(v-v_{0}\right)}^{2}\right]\\ v+k(v-v_{0})\left[{\left(u-u_{0}\right)}^{2}+{\left(v-v_{0}\right)}^{2}\right] \end{array}\right]=H\tilde{P}$$

The translation vector $\hat{t}$ in Equation (1) needs to be further corrected. The expression of the actual microscope magnification *M* and the actual translation vector $\hat{t}$ is obtained below:(5)$$\left\{\begin{array}{c} M=\frac{M_{u}+M_{v}}{2}=\frac{f_{x}*L_{u}+f_{y}*L_{v}}{2\lambda }\\ \hat{t}=L_{u}t=L_{v}t \end{array}\right.$$
where $M_{u}$ and $M_{v}$ represent the magnification of the microscope along the *u* axis and the *v* axis, respectively.

$L_{u}$ and $L_{v}$ represent the physical side lengths of the camera pixel unit along the *u* axis and the *v* axis, respectively.

The transformation matrix caused by the lens distortion is:(6)$$T_{D}=M^{\prime }R\hat{t}$$
where $M^{\prime }$ is the ratio of the actual magnification and the theoretical amplification multiple.

### 3.2. Camera Installation Error

The camera of the micromanipulation system is fixed to the microscope through a screw connection. However, this mechanical fixing method, whether it is a thread or other high-precision connection methods, inevitably has a certain deflection angle which is expressed as a clear deviation between the image coordinate system and the physical coordinate system, as shown in Figure 4, after being highly magnified by the microscope.

The coordinate transformation caused by the deflection of the camera can be written as:(7)$$T_{R}=\left[\begin{array}{ccc} 1 & \varepsilon_{y} & 0\\ \varepsilon_{x} & 1 & 0\\ 0 & 0 & 1 \end{array}\right]$$
where $\varepsilon_{x}$ and $\varepsilon_{y}$ are the conversion factors about rotation deflection angles around the *X*-axis and *Y*-axis of the physical coordinate system, respectively.

### 3.3. Mechanical Displacement Error

The X-Y stage of the micromanipulation system generates the displacement error due to the mechanical drive, which will accumulate with the increase of the displacement. So the mechanical displacement error of the system needs to be compensated. The homogeneous transformation matrix of coordinates can be written as:(8)$$T_{M}=\left[\begin{array}{c} {∆}_{x}\\ {∆}_{y}\\ 1 \end{array}\right]$$
where ${∆}_{x}$ and ${∆}_{y}$ are the mechanical displacement errors of X-direction and Y-direction, respectively.

## 4. Establishment of Error Compensation Model

It can be seen from the last section that the error of the micromanipulation platform is mainly composed of three parts: the image distortion error, the deflection angle error between the image coordinate system and the physical coordinate system, and the mechanical displacement error. So, the compensation for the systematic error mainly compensates for these three parts.

### 4.1. The Compensation Principles

A. The compensation principle of the image distortion error

This article calculates the image distortion error with the standard ruler shown in Figure 5. The parameters required for image distortion error models are obtained through image processing. Image distortion error compensation is mainly implemented through the four steps: automation center line extraction, distortion correction based on linear projection characteristics, estimation of the homography matrix, and the complete solution of internal and external parameters. The specific process and implementation method are shown in Figure 5.

Image distortion can cause the lines of projection not to be straight. Therefore, the problem of parameter solving in the deformed projection model is the problem of non-linear optimization. If there are n straight lines in the physical coordinate system, after the digital camera imaging, the central line is discrete to several discrete points. The distorted coordinates (${\hat{u}}_{i}^{j}$, ${\hat{v}}_{i}^{j}$)^T^ are mapped through the distortion model Equation (3) to obtain the corresponding theoretical point coordinates. Then, the following nonlinear optimization objective function can be established:(9)$$\begin{array}{l} Ϝ\left(k,u_{0},v_{0},\alpha_{1},\alpha_{2},\ldots \ldots \alpha_{n},\varphi_{1},\varphi_{2},\ldots \ldots \gamma_{n}\right)\\ \;\;\;\;\;\;\;\;\;\;\;\;\;\;\;\;\;\;\;\;\;\;\;\;\;\;\;\;=min\sum_{i=1}^{n}\sum_{j=1}^{m_{i}}{\left({\hat{u}}_{i}^{j}\cos\alpha_{i}+{\hat{v}}_{i}^{j}\sin\alpha_{i}+\varphi_{i}\right)}^{2} \end{array}$$
where $\left\{\begin{array}{c} {\hat{u}}_{i}^{j}=u_{i}^{j}+k(u_{i}^{j}-u_{0}){r_{d}}^{2}\\ {\hat{v}}_{i}^{j}=v_{i}^{j}+k(v_{i}^{j}-v_{0}){r_{d}}^{2} \end{array}\right.$

$m_{i}$ represents the number of discrete points on the theoretical center line.

${\left(u_{i}^{j},v_{i}^{j}\right)}^{T}$ and ${\left({\hat{u}}_{i}^{j},{\hat{v}}_{i}^{j}\right)}^{T}$ represent the theoretical coordinate of the *i*-th point on the *j*-th point of the theoretical center line, respectively.

$\alpha_{i}$ and $\varphi_{i}$ represent the tilt angle and interception of the *i*-th theoretical center line, respectively.

According to the target function in Equation (9) and the central line coordinates extracted from the previous step, the distortion coefficient *K*, main point coordinate *(U*_0_, *V*_0_)*^T^*, and the tilt angle $\alpha_{i}$ and interception $\varphi_{i}$ can be obtained through the Levenberg–Marquardt optimization algorithm.

Since the entire projection is a 2D projection, the homography matrix *H* is a 3 × 3 matrix. For the intersection line pattern, through the above steps, 4 pairs of parallel straight lines and 4 pairs of intersecting dots can be obtained through the above steps. Thus, the complete estimation of the homography matrix is further achieved. Assume that $\left(P_{i}\to p_{i}\right)$ and $\left(L_{i}\to l_{i}\right)$ are the point pairs of the line pairs after distortions. Then, from the Equations (1) and (2), the following can be obtained:(10)$$\left\{\begin{array}{c} P_{i}\times Hp_{i}=0\\ l_{i}\times HL_{i}=0 \end{array}\right.$$

By means of a singular value decomposition matrix, the homologous matrix *H* can be obtained, and $f_{x}$, $f_{y},(u_{0},v_{0}),k,M$, R,t can be further obtained through matrix decomposition.

B. The compensation principle of the deflection error between image coordinate system and physical coordinate system

The deflection angle error between the image coordinate system and the physical coordinate system is primarily caused by the installation error of the camera. For the calculation of the deflection angle between the image coordinate system and the physical coordinate system, a series of frames that the previous frame and the after frame has an 1/2 width’s (while displacing along X-direction of the stage) or height’s (while displacing along Y-direction of the stage) overlap are obtained by using the standard rule shown in Figure 5. Take X-direction displacement as an example: two adjacent frames with partially overlapping images as shown in Figure 6a are selected and processed by the computer for image stitching. In the process, the SIFT algorithm [23] is used for image feature extraction, the RANSAC algorithm [24] is applied for image feature matching to eliminate outliers, and key matching points are put into use for image stitching. Finally, the displacement errors *dy* and *dx* can be calculated using the stitched image as shown in Figure 6b. Then, the deflection angle error θ between the image coordinate system and the physical coordinate system is obtained, i.e., θ of the camera installation is written as
(11)$$\theta =arctan\frac{dy}{dx}$$

C. The compensation principle of the mechanical displacement error

In a certain displacement direction, the difference between the theoretical and the actual displacement value of the platform is the displacement error in this direction after the compensation of the deflection angle error. Thus, the platform was moved several times in a fixed step along the positive and negative directions of the X-direction and the Y-direction, respectively, and the corresponding images were captured by the camera after each displacement. After that, the image stitching algorithm mentioned in Section 4.1B can be used to obtain the average displacement error of the platform along the X-direction and Y-direction which can be represented by ${∆x}_{+}$, ${∆x}_{-}$, ${∆y}_{+}$ and ${∆y}_{-}$, respectively.

Taking the X-direction as an example, a schematic diagram of mechanical displacement error calculation is shown in Figure 7 after the deflection angle error is compensated. Then, the calculation formula of ${∆x}_{+}$ is written as
(12)$${∆x}_{+}=\frac{1}{n}\left({\left({∆x}_{+}\right)}_{12}+{\left({∆x}_{+}\right)}_{23}+\cdots {\left({∆x}_{+}\right)}_{\left(n-1\right)n}+{\left({∆x}_{+}\right)}_{n\left(n+1\right)}\right)\;=\frac{1}{n}\sum_{i=1}^{n}{\left({∆x}_{+}\right)}_{i\left(i+1\right)}$$
where ${\left({∆x}_{+}\right)}_{12}$ is the displacement error value obtained after image stitching of the first image and the second image in the positive direction of X-direction; ${\left({∆x}_{+}\right)}_{23}$ is the displacement error value of the second and third images. Similarly, ${\left({∆x}_{+}\right)}_{n\left(n+1\right)}$ is the displacement error value of the nth and nth+1 images.

Based on the axisymmetric principle, ${∆x}_{-},{∆y}_{+}$ and ${∆y}_{-}$ have similar calculation principles to ${∆x}_{+}$.

When the platform moves in a non-axial direction, the displacement error of the system is combined by the X-direction and Y-direction systematic errors. For example, as shown in Figure 8, when displacing from point *O* to point *A* in the physical coordinate system, the mechanical displacement error consists of ${∆x}_{+}$ and ${∆y}_{+}$. While it is in other quadrants, the corresponding mechanical displacement errors distribution is also shown in Figure 8.

Thus, the formula for calculating the compensation coefficient of mechanical displacement error is modeled as:(13)$$\mu =\left[\begin{array}{c} \begin{array}{cc} 1+{∆x}_{+} & 1+{∆y}_{+}\\ 1+{∆x}_{-} & 1+{∆y}_{+} \end{array}\\ \begin{array}{cc} 1+{∆x}_{-} & 1+{∆y}_{-}\\ 1+{∆x}_{+} & 1+{∆y}_{-} \end{array} \end{array}\right]\left[\begin{array}{c} cos\beta \\ sin\beta \end{array}\right]\;=\left[\begin{array}{c} (1+{∆x}_{+})cos\beta_{1}+(1+{∆y}_{+})sin\beta_{1}\\ (1+{∆x}_{-})cos\beta_{2}+(1+{∆y}_{+})sin\beta_{2}\\ (1+{∆x}_{-})cos\beta_{3}+(1+{∆y}_{-})sin\beta_{3}\\ (1+{∆x}_{+})cos\beta_{4}+(1+{∆y}_{-})sin\beta_{4} \end{array}\right]$$
where μ is the mechanical error compensation coefficient of the micromanipulation platform. β is the positive deviation angle between the displacement direction and the X-direction. $\beta_{1}$ belongs to First quadrant, $\beta_{2}$ belongs to Second quadrant, $\beta_{3}$ belongs to Third quadrant, $\beta_{4}$ belongs to Fourth quadrant.

### 4.2. Error Compensation Model

Combined with Equations (6)–(8), the error transformation matrix can be obtained:(14)$$T=M^{\prime }\left[\begin{array}{ccc} 1 & \varepsilon_{y} & {∆}_{x}\\ \varepsilon_{x} & 1 & {∆}_{y}\\ 0 & 0 & 1 \end{array}\right]R\hat{t}$$

To compensate the system errors, an error compensation model needs to be established. As shown in Figure 9, when the stage is expected to move from point $A(X_{0},Y_{0})$ to point $B(X_{1},Y_{1})$, due to the system error, the stage actually moves to point $B^{\prime }(X_{1}^{\prime },Y_{1}^{\prime })$. This can be expressed as
(15)$$\bar{S}=T\bar{S^{\prime }}$$
where $\bar{S}$ and $\bar{S^{\prime }}$ are the vectors of expected displacement and actual displacement, respectively.

Further,
(16)$$\left\{\begin{array}{c} X_{1}=X_{0}+{M^{\prime }\delta }_{x}s\cos\theta \\ Y_{1}=Y_{0}+M^{\prime }\delta_{y}s\sin\theta \end{array}\right.$$
where $\delta_{x}$ and $\delta_{y}$ are the first and second lines of $R\hat{t}$, which represent the compensation coefficient of the image distortion of the X-direction and the Y-direction, respectively. s is the magnitude of $S^{\prime }$.

Since the mechanical error of X-Y stage is positively linearly correlated with the moving distance, the error compensation model is as follows:(17)$$\left\{\begin{array}{c} X_{1}=X_{0}+{M^{\prime }\delta }_{x}\mu *s\cos\theta \\ Y_{1}=Y_{0}+M^{\prime }\delta_{y}\mu *s\sin\theta \end{array}\right.$$

It can be seen from Equation (17) that the establishment of the error model is mainly related to the five compensation coefficients $M^{\prime }$, $\delta_{x}$, $\delta_{y}$, μ and θ.

## 5. Experimental Results and Discussion

### 5.1. Calibration of Error Compensation Model

In order to determine the error compensation formula, an experiment was designed in this paper. A high-precision scale, the center of which consists of 20 × 20 squares with a side length of 50 μm, was selected as the standard ruler, and the image resolution was set as 1200 × 900. When the image samples were collected, the X-Y stage was used to adjust the step length under a 20× objective so that there is an overlap of 1/2 between adjacent images. A total of 40 images were taken according to the stepping method shown in Figure 10.

The error compensation coefficients of image distortion are obtained through the algorithm of Section 4.1A. The results are shown in Table 1.

Among the 40 images, 10 groups of pairwise adjacent images were randomly selected for image stitching to obtain the error compensation coefficient of the deflection error between image coordinate system and platform coordinate system. Image stitching was performed on adjacent images. The effect diagram of this when moving in the X-direction is shown in Figure 11a, and when moving in the Y-direction is shown in Figure 11b. Thus, the deflection angle error compensation coefficient θ is obtained. The results are shown in Table 2.

The mechanical displacement error obtained by image stitching is shown in Table 3.

Thus, the compensation coefficient of mechanical displacement error can be obtained.
(18)$$\mu =-\left[\begin{array}{c} 4.2cos\beta_{1}+4.7sin\beta_{1}\\ 4.8cos\beta_{2}+4.7sin\beta_{2}\\ 4.8cos\beta_{3}+4.2sin\beta_{3}\\ 4.2cos\beta_{4}+4.2sin\beta_{4} \end{array}\right]$$

According to Equations (17), the error compensation formula of the micromanipulation platform is further obtained as follows:(19)$$\left\{\begin{array}{c} X_{1}=X_{0}-2.341*\left[\begin{array}{c} 4.2cos\beta_{1}+4.7sin\beta_{1}\\ 4.8cos\beta_{2}+4.7sin\beta_{2}\\ 4.8cos\beta_{3}+4.2sin\beta_{3}\\ 4.2cos\beta_{4}+4.2sin\beta_{4} \end{array}\right]*s)\\ Y_{1}=Y_{0}-0.264*\left[\begin{array}{c} 4.2cos\beta_{1}+4.7sin\beta_{1}\\ 4.8cos\beta_{2}+4.7sin\beta_{2}\\ 4.8cos\beta_{3}+4.2sin\beta_{3}\\ 4.2cos\beta_{4}+4.2sin\beta_{4} \end{array}\right]*s) \end{array}\right.$$

### 5.2. Single Shot Error Test

In order to further measure the accuracy of the error compensation formula of the micromanipulation platform, the single shot error test was performed in this paper, and the center of the image was set as the origin. The test was started by clicking the mouse to randomly select points of interest other than the origin. Then, the relative distance size between the selected point of interest and the origin in the image coordinate system was calculated, and the corresponding step command was transferred to the microoperation platform, so that the point of interest displaces to the origin. Finally, the position error between the point of interest after displacing and the origin can be calculated as the single shot error, which is also the error value of the algorithm after compensation. The single shot error test interface is shown in Figure 12.

Based on the above-mentioned single shot error test rule, 50 groups of tests before and after the error compensation were carried out, respectively. The results for the absolute values of the difference in X-direction and Y-direction between the point of interest after displacing and the target position (the origin of the coordinate system in the diagram) are plotted in the coordinate system shown in each figure of Figure 13.

The comparison of the test results shows that the errors of the single shot error test before compensation are all above 0.7 μm, and some even as high as 1 μm. But after compensation, the error of the single shot error test is basically within 0.2 μm, the coverage rate is 82%, the maximum error is within 0.25 μm, and the errors are all within the acceptable range of biological operations.

### 5.3. Cumulative Error Test

The experimental analysis using the single shot error test has proved that the systematic error of the micromanipulation platform can be basically eliminated by the method in this paper. However, the single shot error test is only a test and analysis of the error compensation effect in a single displacement direction, which cannot reflect the cumulative error after multiple displacements. Therefore, in order to further measure the cumulative errors after the error compensation of the micromanipulation system, the following experiments were designed for analysis. First, image sample collection was carried out through the visual module using the image acquisition method shown in Figure 11. By adopting this method, if there is no systematic error, the first image and the last image of each group should line up over each other exactly in theory. Therefore, the image stitching error between the first and last image can be calculated as the cumulative error value.

According to the above methods, 50 groups of image samples before and after error compensation were collected. The classification of the 50 groups of experiments is shown in Table 4.

According to the image stitching error of samples, the cumulative error distribution diagram was obtained as shown in Figure 14. Before compensation, the displacement error in the X-direction of the 50 groups was basically in the range of 1~1.4 μm, and which in the Y-direction was −2.9~−4.8 μm. After the error compensation, the displacement error in the X-direction of the 50 groups of samples was reduced to 0.01~0.06 μm, and that in the Y-direction was reduced to −0.02~−0.07 μm. In addition, through five kinds of experiments, it can be seen that as the total step length in each direction increased, the cumulative error in both the X-direction and the Y-direction also increased. However, before compensation, the overall X-direction error difference between classification 5 and classification 1 shown in Table 4 is about 0.4 μm, and the Y-direction error difference is about 1.9 μm. In comparison, the compensated X-direction error difference can be controlled within 0.05 μm, and the Y-direction can be controlled within 0.07 μm. For every 1000 μm of platform displacement, the cumulative error increases within 0.02 μm. It can be seen that through the error compensation, the cumulative error of the micromanipulation platform has been significantly reduced, so that its error value can meet the basic accuracy requirements of the micromanipulation.

## 6. Conclusions

To reduce positioning errors in micromanipulation, modeling and compensation principles were proposed to deal with the error caused by the non-linear imaging distortion and mechanical errors such as camera installation error and the mechanical displacement error of the micromanipulation platform. The influences and derivation of those errors are analyzed first. Then, a novel and comprehensive error compensation model is established based on the compensation principle of each error. The distortion error compensation coefficients were obtained by the Levenberg–Marquardt optimization algorithm combined with the deduced nonlinear imaging model. The mechanical error compensation coefficients were derived from the rigid-body translation technique and image stitching algorithm. Finally, to validate the error compensation model, single shot and cumulative error tests were designed. The experimental results show that the positioning accuracy of the micromanipulation system has been significantly improved. The systematic error of a single displacement can be controlled within 0.25 μm, while the cumulative displacement is controlled within 0.02 μm per 1000 μm, which basically meets the positioning requirements of cell microorganisms.

## Figures and Tables

**Figure 1 micromachines-14-00779-f001:**
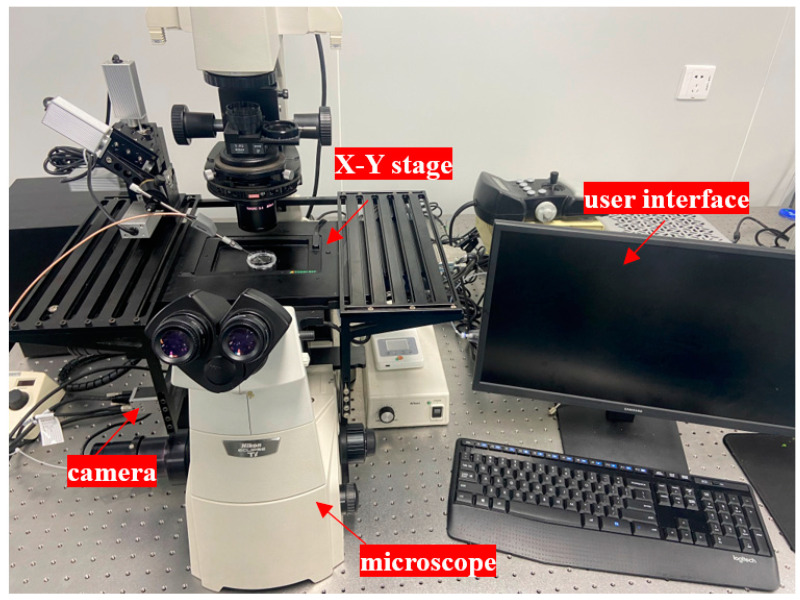
The hardware system of the micromanipulation system.

**Figure 2 micromachines-14-00779-f002:**
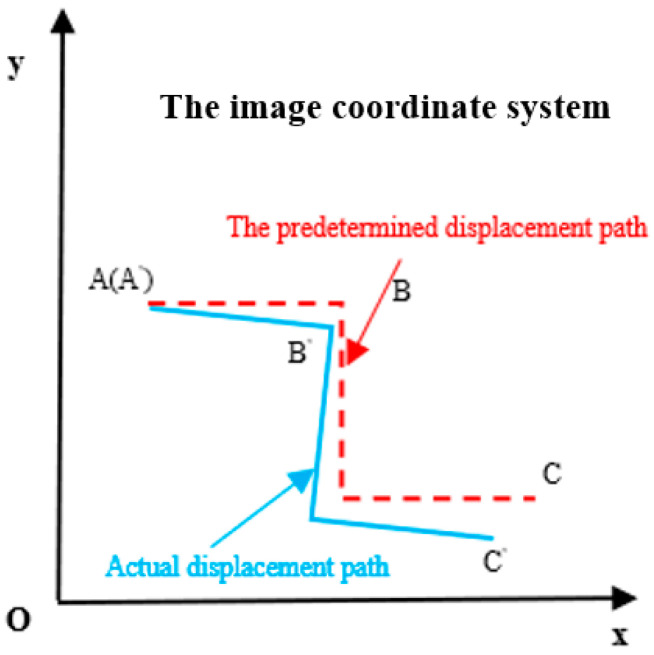
Deviation of displacement path caused by system error. The predetermined displacement path is moved from point A to point C via point B, while the actual displacement path of the platform is moved from point A′ to point C′ via B′ which deviates from the predetermined displacement path due to the system errors.

**Figure 3 micromachines-14-00779-f003:**
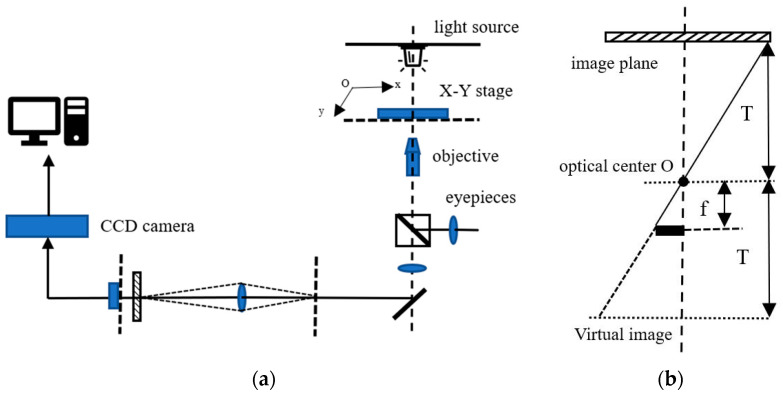
The vision system and simplified imaging model of the micromanipulation system. (**a**) The diagram of the vision system, (**b**) The simplified imaging model. *T* is the working distance of the microscope and *f* is the focal length of the objective lens.

**Figure 4 micromachines-14-00779-f004:**
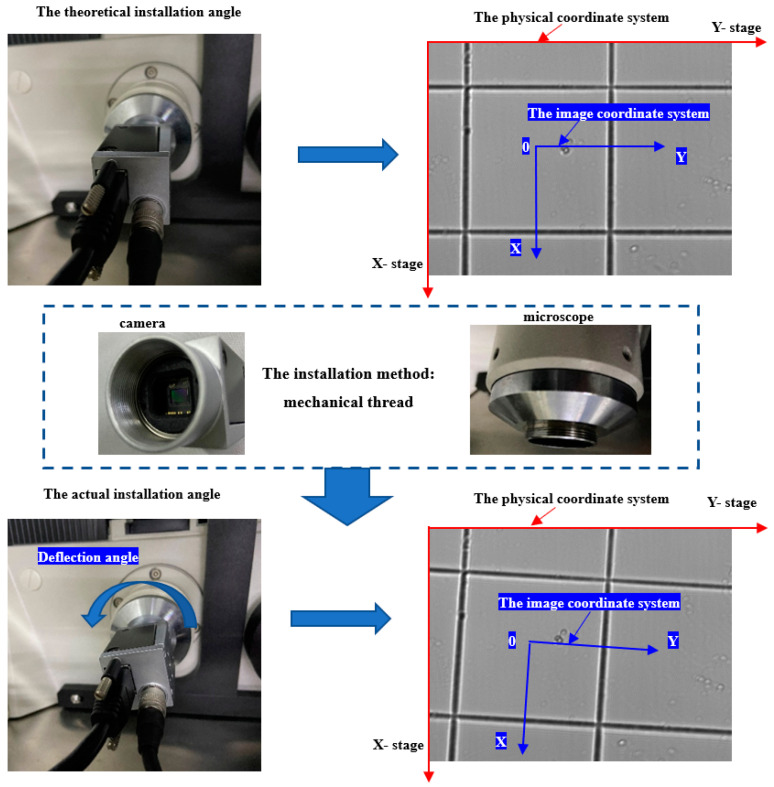
Deviation of the image coordinate caused by camera deflection angle. The camera is connected with the microscope by mechanical threaded. This installation method is easy to produce a deflection angle which is expressed as a clear deviation between the image coordinate system and the physical coordinate system.

**Figure 5 micromachines-14-00779-f005:**
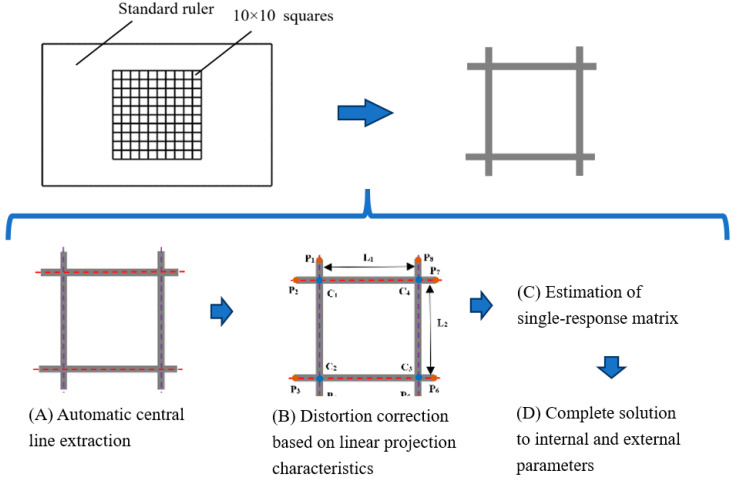
The process of image distortion error.

**Figure 6 micromachines-14-00779-f006:**
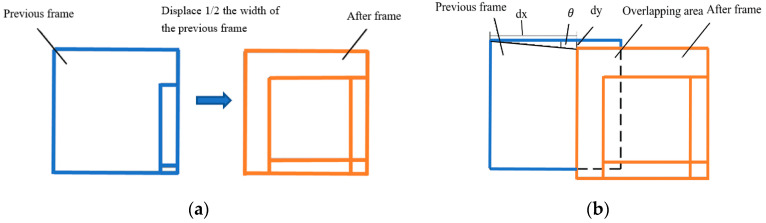
Image stitching schematic diagram. (**a**) Two adjacent images, (**b**) Image stitching.

**Figure 7 micromachines-14-00779-f007:**
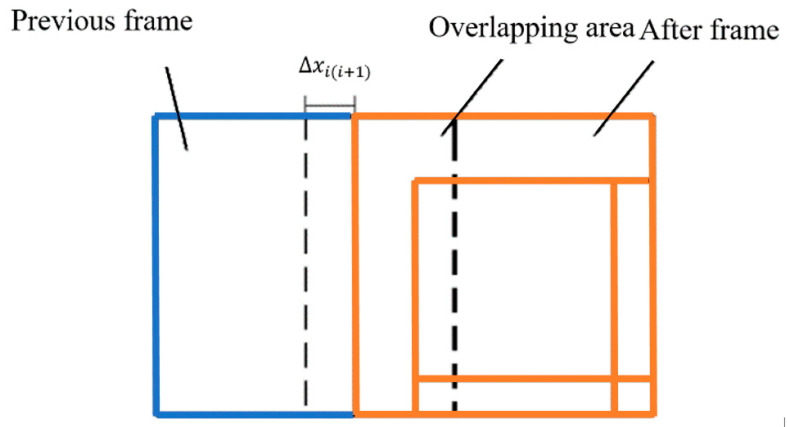
Schematic diagram of mechanical displacement error calculation.

**Figure 8 micromachines-14-00779-f008:**
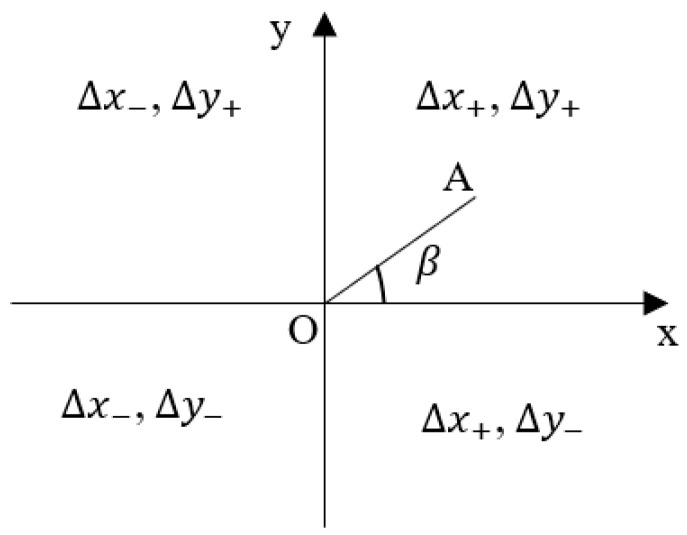
The distribution of mechanical displacement error.

**Figure 9 micromachines-14-00779-f009:**
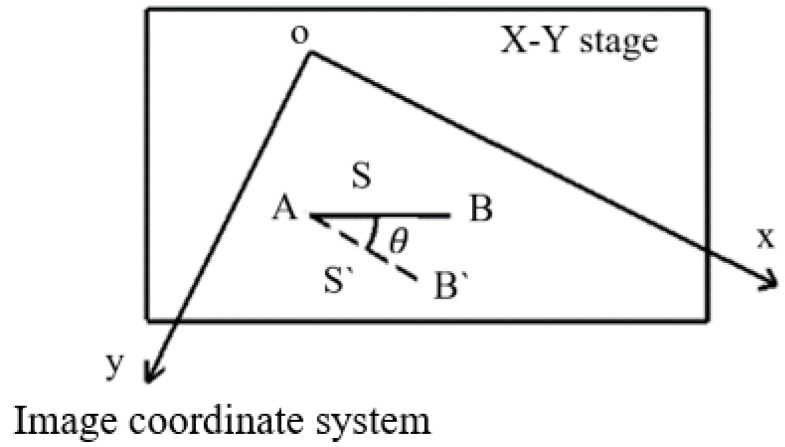
Schematic diagram of expected displacement and actual displacement. The expected vector displacement $\bar{S}$ is from point *A* to point *B*, but the actual displacement becomes vector $\bar{S^{\prime }}$ that is from point *A* to point $B^{\prime }$. The deflection angle between the expected displacement and actual displacement is θ.

**Figure 10 micromachines-14-00779-f010:**
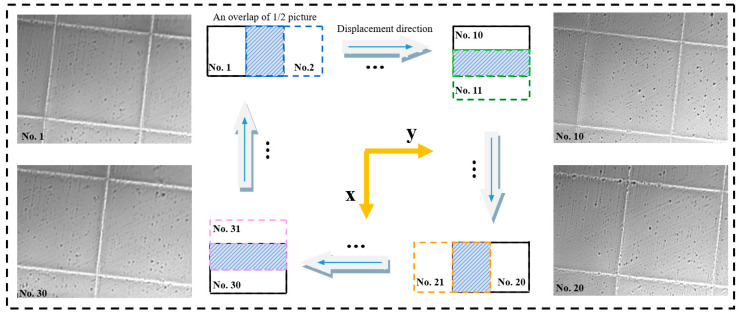
Image acquisition method. A total of 40 pictures were collected over multiple displacements by using the X-Y stage with a step size of half the width of the image (when it is displaced along the X-direction) or half the height (when it is displaced along the Y-direction). No.1 represents the first picture. No.2 represents the second picture, and so on.

**Figure 11 micromachines-14-00779-f011:**
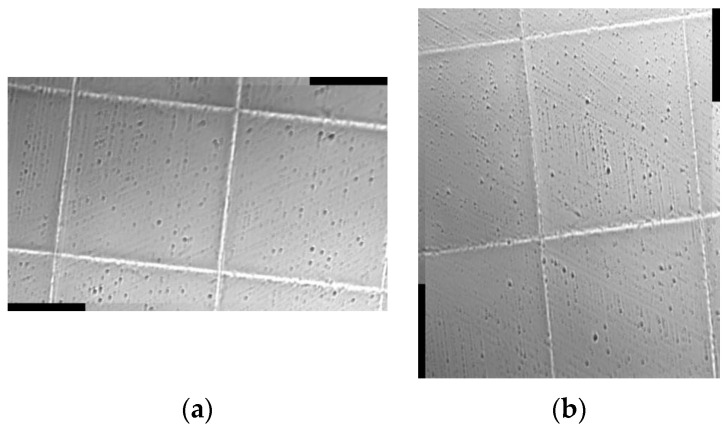
Effect diagram of image stitching. (**a**) X-direction, (**b**) Y-direction.

**Figure 12 micromachines-14-00779-f012:**
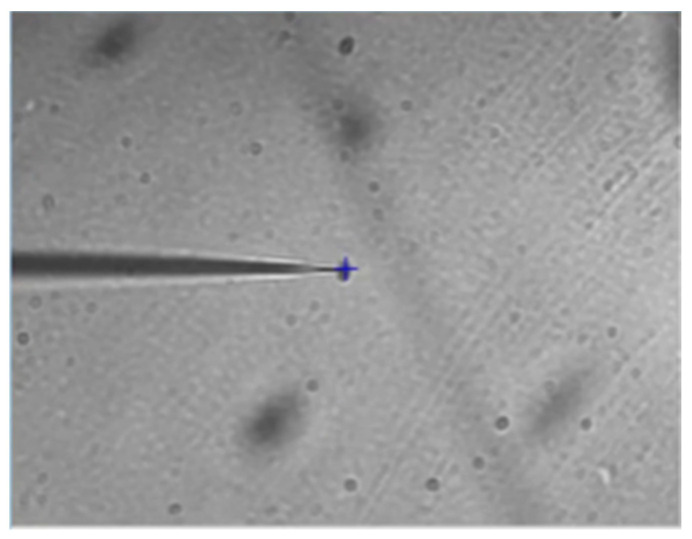
Single shot error test interface.

**Figure 13 micromachines-14-00779-f013:**
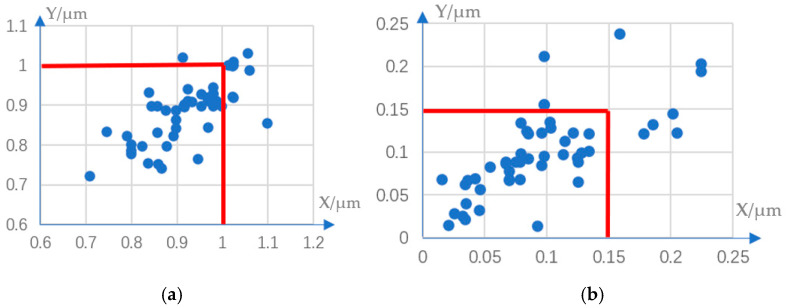
Single shot error before and after compensation. (**a**) before, (**b**) after.

**Figure 14 micromachines-14-00779-f014:**
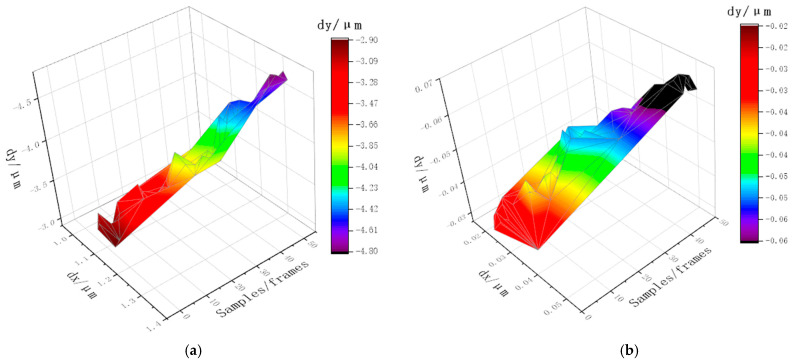
Cumulative errors before and after compensation. (**a**) before, (**b**) after.

**Table 1 micromachines-14-00779-t001:** Results of error compensation coefficient *M*, $\delta_{x}$ $\delta_{y}$.

Parameters	Theoretical Values	Actual Values
$f_{x}$ (pixel/µm)	1	0.993
$f_{y}$ (pixel/µm)	1	0.992
$\left(u_{0},v_{0}\right)$ (pixel)	(600.0,450.0)	(600.3,450.2)
k (pixel)^−2^	0	0.0122
M	20	19.843
R	$\left[\begin{array}{ccc} 1.000 & 0 & 0\\ 0 & 1.000 & 0\\ 0 & 0 & 1.000 \end{array}\right]$	$\left[\begin{array}{ccc} 0.999 & -0.045 & 0.021\\ 0.02 & 1.000 & 0.013\\ -0.014 & -0.05 & 0.999 \end{array}\right]$
t	${\left[\begin{array}{ccc} 0 & 0 & 0 \end{array}\right]}^{T}$	${\left[\begin{array}{ccc} -2.5 & -2.7 & 0.2 \end{array}\right]}^{T}$
$M^{\prime }$	1	0.9922
$\delta_{x}$	0	−2.3718
$\delta_{y}$	0	−2.7474

**Table 2 micromachines-14-00779-t002:** Results of error compensation coefficient θ.

Groups	Result of Error Compensation Coefficient θ		
	dy (pix)	dx (pix)	θ
1	−30	304	−5.636
2	−29	306	−5.414
3	−30	308	−5.563
4	−31	308	−5.747
5	−29	303	−5.467
6	−30	307	−5.581
7	−30	306	−5.599
8	−31	308	−5.747
9	−29	305	−5.431
10	−30	306	−5.599
average	-	-	−5.578

**Table 3 micromachines-14-00779-t003:** Results of ${∆x}_{+}$, ${∆x}_{-}$, ${∆y}_{+}$ and ${∆y}_{-}$.

Groups	${∆x}_{+}$ (pix)	${∆x}_{-}$ (pix)	${∆y}_{+}$ (pix)	${∆y}_{-}$ (pix)
1	−3	−4	−6	−4
2	−7	−6	−6	−5
3	−6	−8	−6	−5
4	−5	−8	−5	−7
5	−5	−6	−5	−6
6	−4	−5	−7	−5
7	−3	−7	−5	−4
8	−6	−5	−6	−5
9	−7	−4	−5	−6
10	−6	−5	−6	−5
average	−5.2	−5.8	−5.7	−5.2

**Table 4 micromachines-14-00779-t004:** Classification of the experiments.

Classification	Groups	The Step Length (μm)	Total Number of Images in a Group	Number of Displacement Images in One Direction	Total Step Length in One Direction (μm)
1	1–10	600	5	2	1800
2	11–20	600	9	3	2400
3	21–30	600	13	4	3000
4	31–40	600	17	5	3600
5	41–50	600	21	6	4200

## Data Availability

Not applicable.

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
