# Peer review of "Modeling and Compensation of Positioning Error in Micromanipulation"

_micromachines, 2023, doi:10.3390/mi14040779_

Round 1

Reviewer 1 Report

This paper proposed a novel error compensation method with consideration of the microscope nonlinear imaging distortion, camera installation error, and the mechanical displacement error of the motorized stage to improve the positioning accuracy of the micromanipulation system. The method is interesting, but there are still many problems in the manuscript. My specific comments are as follows. 

  1. The summary of related work is lacking, and it is difficult to understand where the innovations and contributions of this paper are.
  2. There are many problems with the formatting and layout of the article. For instance, the units in Fig. 13 are missing; Tables 1 and 3 are both separated into two pages; Page 13 has a large blank space.
  3. The text has many errors and needs to be checked carefully. For example, the first paragraph of Section 1 ends with ","; the first paragraph of Section 3.1 ends without "."; "Where:"; Some variables are in italics, but some are not; "" should not be used in English papers; etc.

Reviewer 2 Report

I suggest this manuscript can be published after the major revisions according to the above suggestions.

Reviewer 3 Report

In this paper, a comprehensive error model is established, and a new error compensation method is proposed and verified. The structure of the paper is clear, however, it still needs some minor revisions:

1. The pictures of some small figures are not left aligned with the figure numbers, such as Fig. 11, Fig.12, etc

2. The picture and the figure number are paginated.

3. Pages 4, 6 and 13 all have blank lines of varying degrees.

4. Some sentences have grammatical flaws.

5. the conclusion should be reworked with more clear description of the main innovation of your work.

6. Fig.3, the left should be (a) and right (b).

Round 2

Reviewer 1 Report

Please carefully check the paper again, there are minor errors, such as the last author's working unit is missing, or you make a wrong number.

Author Response

Dear reviewer,

I revised the draft and thank you to allow me to resubmit the revised version. Here are the detailed changes in the new version.

1. Re-checked and corrected the mistakes of the author and the author's working unit.
2. Fixed other minor errors. For example, references [22], [21], [24] were forgotten to be revised in the previous version, and they are still marked as [19], [20], [21] in the main text.

Reviewer 2 Report

本文详细分析了显微镜的一些误差,并提出了相应的误差补偿模型。设计相应的实验对模型进行验证,结果是可以接受的。经过作者的修改,文章已经比较完善了。稿件可以接受。

Author Response

(The authors gave the same response as above.)
